# Comprehensive Evaluation of Low Nitrogen Tolerance in Oat (*Avena sativa* L.) Seedlings

**Yue Wang** [1]**, Kaiqiang Liu** [1]**, Guoling Liang** [1]**, Zhifeng Jia** [1]**, Zeliang Ju** [1] **, Xiang Ma** [1] **and Qingping Zhou** [1,2,]*****

[1] Key Laboratory of Superior Forage Germplasm in the Qinghai-Tibetan Plateau, Qinghai Academy of Animal Husbandry and Veterinary Sciences, Qinghai University, Xining 810016, China
[2] Sichuan Zoige Alpine Wetland Ecosystem National Observation and Research Station, Southwest Minzu University, Chengdu 610041, China
***** Correspondence: qpingzh@aliyun.com; Tel.: +86-158-0285-6077

**Abstract:** In oat production, the over-application of nitrogen (N) fertilizer in fields due to low N fertilizer use efficiency not only increases production costs but also causes environmental pollution. Currently, mining low N-tolerant oat varieties is an important way to promote sustainable agriculture. In this study, 30 oat varieties were grown in a seedling culture with two treatments of normal N (10 mM $NH_4NO_3$) and low N (1.25 mM $NH_4NO_3$), and the correlations between agronomic traits and plant N content and low N tolerance coefficients and indices were determined, which can be used as indicators for the evaluation of low N-tolerant oat varieties. Coefficient of variation, correlation analysis, principal component analysis, partial least-squares discrimination analysis, random forest analysis, least absolute shrinkage and selection operator regression and model evaluation, and membership function analysis were used for in-depth analysis of these indicators. Plant N content, root–crown ratio, and dry weight of aboveground plant parts were found to be important indicators of low N tolerance in oats. According to the membership function ranking of the 30 selected oat varieties, Jiayan 2, Qingyongjiu 035, and Qingyin 2 had strong tolerance to low N stress and Qingyongjiu 003, Qingyongjiu 021, and Qingyongjiu 016 had poor tolerance to low N stress. Thus, this study provides a reliable and comprehensive method for evaluating the low N tolerance of oat varieties as well as a reference for screening other low N-tolerant plants.

**Keywords:** oats; low nitrogen tolerance; evaluation index; screening

## 1. Introduction

Oat (*Avena sativa* L.), which belongs to the family Poaceae, is a forage annual herbaceous crop with a wide distribution and is an indispensable crop in fragile ecological zones [1,2]. Compared to other crops, oats are resistant to aridity, salinity, drought, and cold [3]. Oats can be diploid, tetraploid, and hexaploid, with oat cultivars being heterozygous hexaploids [2]. Proteins, carbohydrates, and trace elements in this irreplaceable forage crop can enhance the meat quality of cattle and sheep [4,5]. The food crop is highly cultivated and consumed for its high nutritional composition, health functions, and medicinal value [6].

Nitrogen (N) is one of the three macronutrients, along with phosphorus and potassium, that are essential for plant growth and development [7]. Thus, a low N level is a major limiting factor for plant growth and yield [8,9]. The rate of N fertilizer application has been widely demonstrated as a key yield determinant in oat production [10]. With about 30% of global N fertilizer application, the use of N fertilizer by China is the highest, which is much higher than the world average [11]. However, grain production is low when N fertilizer is overused, resulting in excessive N pollution that threatens environmental security and sustainable agriculture, as N fertilizer or nitrate is a major source of water pollution [12–14]. To meet N fertilizer demand and minimize environmental pollution, there is a need to improve N fertilizer use efficiency in crops [15].

It is important to reduce the use of chemical fertilizers, increase crop production efficiency, prioritize ecological preservation, promote high-quality development, create a high-quality life, and develop ecological agriculture in the crop cultivation regions. In actual production, to minimize the use of N fertilizer without affecting the crop yield, measures such as choosing appropriate fertilization methods and screening of low N-tolerant varieties can be used. Among these, N fertilizer depth application significantly improves N recovery efficiency compared with N spreading application [16]. In addition, the early farmers' quest for high yields and excessive use of chemical fertilizers meant that there was less research into long-term pollution levels of nitrogen fertilizers and low nitrogen breeding in early breeding objectives. It is clear that bottlenecks to crop yield increases have been reached through traditional excessive fertilization and changes in fertilization methods. Therefore, screening and breeding of low N-tolerant crops can effectively reduce the application of chemical fertilizers, decrease agricultural production costs, and protect the environment. At present, low N-tolerant varieties of plants such as rice [17], wheat [18], sorghum [19], and maize [20] have been identified by screening in hydroponic trials. However, research on the screening of oat varieties for low N tolerance is lacking. Therefore, in this study, 30 oat varieties were selected, and 10 agronomic traits, as well as the plant N content, were evaluated in oat seedlings under normal and low N treatments using hydroponics. From our results, we found that the plant N content, root–crown ratio, and dry weight of aboveground plant parts are important indicators of the ability of oats to tolerate low N. A comprehensive analysis of the membership function was used to select three strong low N tolerance oat varieties and three poor low N tolerance oat varieties, thus providing a research basis for cultivating oats with low N tolerance.

## 2. Materials and Methods

### 2.1. Test Materials

Thirty oat varieties were provided by the Key Laboratory of Superior Forage Germplasm in the Qinghai-Tibetan Plateau, Qinghai Academy of Animal Husbandry and Veterinary Sciences, Qinghai University, Xining, China (Table 1).

**Table 1.** Names and sources of 30 oat varieties.

| Number | Name | Source | Number | Name | Source |
|--------|------|--------|--------|------|--------|
| 1 | Qingyongjiu 086 | Switzerland | 16 | Qingyongjiu 044 | Canada |
| 2 | Qingyongjiu 068 | Hungary | 17 | Qinghai 444 | China |
| 3 | Qingyongjiu 028 | Soviet Union | 18 | Qingyin 2 | China |
| 4 | Qingyongjiu 021 | China | 19 | Qingyan 1 | China |
| 5 | Qingyongjiu 067 | China | 20 | Qingyongjiu 872 | China |
| 6 | Qingyongjiu 016 | China | 21 | Qingyongjiu 112 | China |
| 7 | Qingyongjiu 065 | Soviet Union | 22 | Qingyongjiu 097 | Sweden |
| 8 | Qingyongjiu 008 | China | 23 | Qingyongjiu 087 | Hungary |
| 9 | Qingyongjiu 055 | Romania | 24 | Qingyongjiu 091 | Hungary |
| 10 | Qingyongjiu 003 | China | 25 | Qinghaitianyanmai | China |
| 11 | Qingyongjiu 045 | Canada | 26 | Jiayan 2 | China |
| 12 | Qingyongjiu 035 | Canada | 27 | Qingyongjiu 093 | Hungary |
| 13 | Bayan 3 | China | 28 | Linna | China |
| 14 | Bayan 5 | China | 29 | Qingyongjiu 096 | Sweden |
| 15 | Qingyongjiu 002 | China | 30 | D1 | China |

### 2.2. Trial Design

The hydroponic experiments were performed in an artificial climate chamber at the Qinghai Academy of Animal Science and Veterinary Medicine on 11 May 2021. An appropriate amount of uniform-sized oat seeds were disinfected with a 2% NaOCl (sodium hypochlorite) solution for 10 min, rinsed well with water, immersed in distilled water overnight at 4 °C, and placed on moist filter paper until germination. Six days later, oat seedlings of uniform growth were selected, wrapped with sponge strips, placed in a

floating nursery, and transferred to perforated foam boards. Each foam board contained 48 holes (6 × 8) on average, and 1 plant was planted in each hole. Afterward, the oat seedlings were placed in a black box (size 60 cm × 50 cm × 16 cm) containing a modified Hoagland's nutrient solution (Normal N (as control) 10 mM $NH_4NO_3$ (ammonium nitrate)/low N 1.25 mM $NH_4NO_3$, 945 mg/L $Ca(NO_3)_2 \cdot 4H_2O$ (calcium nitrate tetrahydrate), 506 mg/L $KNO_3$ (potassium nitrate), 136 mg/L $KH_2PO_4$ (potassium dihydrogen phosphate), 493 mg/L $MgSO_4$ (magnesium sulfate), 2.5 mL/L iron salt solution, and 5 mL/L trace element solution). The components of the iron salt solution were 2.78 g $FeSO_4 \cdot 7H_2O$ (ferrous sulfate heptahydrate), 3.73 g EDTA–2Na (Ethylenediaminetetracetic acid disodium), and 500 mL distilled water. The components of the trace element solution were 0.83 mL/L KI (potassium iodide), 6.2 mg/L $H_3BO_3$ (boric acid), 22.3 mg/L $MnSO_4$ (manganese sulfate), 8.6 mg/L $ZnSO_4$ (zinc sulfate), 0.25 mg/L $Na_2MoO_4$ (sodium molybdate), 0.025 mg/L $CuSO_4$ (copper sulfate), and 0.025 mg/L $CoCl_2$ (cobalt chloride). The experimental temperature was controlled at 25 °C, the light was provided at 16 h/8 h day and night, the culture was aerated by an aerator pump at 1-h intervals, and the nutrient solution was changed at 5-day intervals. Each treatment was replicated three times. The oat varieties were treated with low N when they reached the three-leaf stage, and the seedlings were collected after 25 days. Plant length, plant height, and root length were measured with a straightedge; plant fresh weight and fresh weights of aboveground and belowground plant parts were weighed on scales. The roots and aboveground plant parts were opened and bagged, killed at 105 °C for 30 min, and dried at 65 °C to a constant mass. Dry weights of aboveground and belowground plant parts were weighed on scales.

Total plant N content was determined by the Kjeldahl method [21]. Each sample was ground and passed through a 0.25-mm sieve, and about 0.1 g was taken for determination. The volume was then fixed at 100 mL, and about 10 mL of the sample was taken for nitrogen determination by a flow analyzer. Three replicates were set up for each treatment per variety.

### 2.3. Data Processing and Calculation Methods

Agronomic traits and plant N content of 30 oat varieties were detected between the different growth conditions (normal and low N). The parameter test and coefficient of variation (CV) were used to describe data distribution by SPSS v26.0 (Umetrics, Umea, Sweden). First, principal component analysis (PCA) was used to preliminarily observe the separate trends between groups in SPSS. The supervision methods, including partial least-squares discrimination analysis (PLS-DA), random forest analysis, and least absolute shrinkage and selection operator (LASSO), were performed in RStudio v4.0 (RStudio, Boston, MA, USA) to identify the crucial growth indices related to low N stress. Based on these algorithms, the machine learning metrics were calculated to obtain variable importance in the projection (VIP), importance score, and LASSO coefficient. By combining these metrics, the performance of the final panel, along with the growth index, was evaluated by receiver operating characteristic (ROC) analysis, and the results were visualized with the "ggplot2" package in RStudio.

The relative index was calculated using the following equation.

Low N tolerance index = low N treatment value/normal treatment value.

LASSO regression analysis: The LASSO regression model was constructed using different oat agronomic traits and plant N content index shapes as inputs and N supply conditions as ending variables. The sample error values and the sum of squares were first recorded using cross-validation, and the value of λ with the smallest error was determined as the best penalty coefficient. The number of intersection points under the corresponding position in the LASSO regression step was found according to this penalty coefficient,

which was the final number of variables included in the model, and the vertical coordinate of the intersection point was the regression coefficient of the variable.

$$\beta = \mathrm{argmin}\left\{\sum_{j=1}^{n}\left[(y_j - \sum_{i=1}^{m}\beta_i X_{ij})^2 + \lambda\sum_{i=1}^{m}\beta_i\right]\right\}$$

Given a data set D $(X, y)$, where $X$ represents the independent variable and $y$ represents the dependent variable. The data are predicted using the established LASSO regression model and parameters, and the output is the probability of the predicted group.

In the membership function analysis, the inverse affiliation function $\mu(X_j)$ denotes the value of the affiliation function of the $\mu(X_j)$ composite indicator, $Xj$ denotes the $\mu(X_j)$ composite indicator value, $X_{max}$ denotes the maximum value of the jth composite indicator, and $X_{min}$ denotes the minimum value of the $\mu(X_j)$ composite indicator.

$$\mu(X_j) = (X_j - X_{\min})/(X_{\max} - X_{\min})j = 1, 2, 3, \dots, n$$

$$\mu(X_j) = 1 - (X_j - X_{\min})/(X_{\max} - X_{\min})j = 1, 2, 3, \dots, n$$

## 3. Results

### 3.1. CV of Each Trait and Plant N Content of Oat Varieties at the Seedling Stage under Different N Supply Conditions

The agronomic traits and plant N content of oat varieties significantly differed between the normal and low N treatments ($p < 0.01$; Table 2). Eleven traits related to N fertilizer use efficiency showed large differences in range, mean, and CV at both N levels, plant length, root length, plant fresh weight, fresh weight of aboveground plant parts, fresh weight of belowground plant parts, plant dry weight, dry weight of aboveground plant parts, dry weight of belowground plant parts, and plant N content increased with increasing N level. Among these, CV among oat varieties was greater than 30% for all traits except for plant length, root length, plant height, and plant N content at both levels of N fertilizer application, and the variation in dry weight of aboveground plant parts was the greatest, which was beneficial to show the differences among germplasm resources. For example, the variation in dry weight of aboveground plant parts ranged from 0.42 to 2.71 g under normal N treatment, with an average of 1.17 g and CV of 46.15%; the variation ranged from 0.12 to 1.19 g under low N treatment, with an average of 0.8 g and CV of 38.75%; and the variation in the low N tolerance index ranged from 0.12 to 2.71 g, with an average of 0.68 g and CV of 48.31.

**Table 2.** Range in agronomic traits and plant N content of oat seedlings under normal and low N treatments.

| Trait | Normal N | | | Low N | | | Low N Tolerance Indices | | |
|---|---|---|---|---|---|---|---|---|---|
| | Range | Mean | CV/% | Range | Mean | CV/% | Range | Mean | CV/% |
| Plant length (cm) | 84.5~137.3 | 105.55 A | 11.32 | 59.7~139.6 | 99.73 A | 16.93 | 59.7~139.6 | 0.94 | 14.97 |
| Plant height (cm) | 48.1~84.7 | 64.74 A | 11.88 | 35.4~80 | 58.23 B | 15.03 | 35.4~84.7 | 0.90 | 14.40 |
| Root length (cm) | 25.6~70.4 | 40.97 A | 20.84 | 17.6~72.2 | 41.57 A | 30.89 | 17.6~72.2 | 1.01 | 26.53 |
| Plant fresh weight (g) | 3.95~28.17 | 12.47 A | 41.30 | 2.58~17.5 | 9.01 B | 31.63 | 2.58~28.17 | 0.72 | 41.80 |
| Fresh weight of aboveground plant parts (g) | 3.24~18.37 | 8.33 A | 42.86 | 1.02~11.51 | 5.72 B | 34.79 | 1.02~18.37 | 0.69 | 45.16 |
| Fresh weight of belowground plant parts (g) | 0.71~9.8 | 4.14 A | 41.55 | 0.8~5.99 | 3.31 B | 30.82 | 0.71~9.8 | 0.80 | 39.97 |
| Dry weight of belowground plant parts (g) | 0.05~0.56 A | 0.32 A | 31.25 | 0.06~0.55 | 0.30 A | 30.00 | 0.05~0.56 | 0.94 | 29.65 |
| Dry weight of aboveground plant parts (g) | 0.42~2.71 | 1.17 A | 46.15 | 0.12~1.91 | 0.80 B | 38.75 | 0.12~2.71 | 0.68 | 48.31 |
| Plant dry weight (g) | 0.47~3.27 | 1.49 A | 41.61 | 0.3~2.33 | 1.11 B | 34.23 | 0.3~3.27 | 0.74 | 42.09 |
| Root–crown ratio (%) | 0.12~0.59 | 0.30 A | 33.33 | 0.17~1.5 | 0.42 B | 35.71 | 0.12~1.5 | 1.4 | 39.14 |
| Plant N content (g/kg) | 33.74~54.58 | 47.98 A | 7.92 | 37.21~51.58 | 44.99 B | 7.98 | 33.74~54.58 | 0.94 | 8.59 |

Note: A and B Means in the same column followed by different alphabets are significantly different ($p < 0.01$).

### 3.2. Pearson's Correlation Coefficient and PCA of Traits of Oat Varieties at Different Levels of N Supply

In this study, correlation analysis was performed on the results of 11 traits (Table 3). Correlations between different traits were observed, with varying extents, and those between most traits reached a highly significant level. Plant dry weight and dry weight of aboveground plant parts were the most correlated, with correlation coefficients reaching 0.989, followed by plant fresh weight, fresh weight of aboveground plant parts, fresh weight of belowground plant parts, dry weight of belowground plant parts, plant height, root–crown ratio, plant length, root length, and plant N content, with correlation coefficients of 0.958, 0.970, 0.771, 0.759, 0.614, 0.579, 0.559, 0.461, and 0.021 and were highly significant under normal and low N levels. Among the trait indicators of oat varieties, the greatest variation was observed in the dry weight of aboveground plant parts, which was negatively correlated with the root-crown ratio, while the remaining 10 indicators were positively correlated with the dry weight of aboveground plant parts. The results of the correlation analysis indicated an appropriate relationship between agronomic traits and plant N content of oat varieties, which helps explain the low N tolerance of oat varieties. PCA was used to comprehensively evaluate the 30 oat varieties for low N tolerance to extract factors to classify the 11 indicators and screen key indicators that represent the low N tolerance of oat varieties. Given that the correlation analysis results could reflect good structural validity among variables (KMO: 0.570, Bartlett's test: <0.001), PCA could be used to characterize the low N tolerance coefficients of different oat varieties for agronomic traits and plant N content. Three principal components (PCs) were extracted using the screening criterion of a characteristic root greater than 1 (Table 4), and the cumulative interpretation rate of the first three PCs was 88.148%. The importance of the different common factor loading coefficients was further calculated to analyze the hidden variables in each principal component. For the main PC1, the factor loadings of plant fresh weight, plant dry weight, fresh weight of aboveground plant parts, and dry weight of aboveground plant parts were larger, indicating that this PC mainly reflected information such as plant material accumulation. For the main PC2, the factor loadings of plant N content, root length, and root-crown ratio were larger, indicating that this PC mainly reflected the response of plant morphology, root-crown ratio, and plant N content to different N concentrations. For the main PC3, the factor loadings of plant length, root length, root-crown ratio, and plant N content were larger, indicating that this PC reflected the greater response of morphology, root-crown ratio, and plant N content to different N concentrations, similar to the response of PC2. Thus, it was concluded that the traits used to evaluate the low N tolerance of oats using single or several traits for evaluating the low N-tolerant oat varieties might not be accurate, and further comprehensive evaluation and screening of multiple traits are needed.

**Table 3.** Pearson's correlation analysis of various traits of oat varieties.

| Trait | Plant Length | Plant Height | Root Length | Plant Fresh Weight | Fresh Weight of Aboveground Plant Parts | Fresh Weight of Belowground Plant Parts | Dry weight of Belowground Plant Parts | Dry Weight of Aboveground Plant Parts | Root-Crown Ratio | Plant N Content | Plant Dry Weight |
|---|---|---|---|---|---|---|---|---|---|---|---|
| Plant length | 1 | | | | | | | | | | |
| Plant height | 0.631 ** | 1 | | | | | | | | | |
| Root length | 0.907 ** | 0.434 * | 1 | | | | | | | | |
| Plant fresh weight | 0.491 ** | 0.629 ** | 0.390 * | 1 | | | | | | | |
| Fresh weight of aboveground plant parts | 0.557 ** | 0.629 ** | 0.461 * | 0.975 ** | 1 | | | | | | |
| Fresh weight of belowground plant parts | 0.259 | 0.518 ** | 0.169 | 0.881 ** | 0.757 ** | 1 | | | | | |
| Dry weight of belowground plant parts | 0.398 * | 0.505 ** | 0.339 | 0.843 ** | 0.775 ** | 0.849 ** | 1 | | | | |
| Dry weight of aboveground plant parts | 0.559 ** | 0.614 ** | 0.461 * | 0.958 ** | 0.970 ** | 0.771 ** | 0.759 ** | 1 | | | |
| Root-crown ratio | −0.327 | −0.412 * | −0.397 * | −0.501 ** | −0.575 ** | −0.262 | −0.084 | −0.579 ** | 1 | | |
| Plant N content | 0.109 | 0.077 | 0.195 | 0.008 | 0.117 | −0.217 | −0.224 | 0.021 | −0.484 ** | 1 | |
| Plant dry weight | 0.551 ** | 0.614 ** | 0.456 * | 0.973 ** | 0.970 ** | 0.815 ** | 0.843 ** | 0.989 ** | −0.498 ** | −0.030 | 1 |

Note: * $p < 0.05$, ** $p < 0.01$.

**Table 4.** Eigenvalues and variance contributions of PCA of various indicators of oat seedlings under normal and low N treatments.

| Factor | Eigenvalues | | | % of Variance (Rotated) | | |
|---|---|---|---|---|---|---|
| | Eigenvalue (Unrotated) | % of Variance | Cumulative % of Variance | Eigenvalue (Unrotated) | % of Variance | Cumulative % of Variance |
| 1 | 6.685 | 60.771 | 60.771 | 6.685 | 60.771 | 60.771 |
| 2 | 1.842 | 16.748 | 77.518 | 1.842 | 16.748 | 77.518 |
| 3 | 1.169 | 10.63 | 88.148 | 1.169 | 10.63 | 88.148 |
| 4 | 0.512 | 4.655 | 92.803 | | | |
| 5 | 0.443 | 4.029 | 96.832 | | | |
| 6 | 0.2 | 1.815 | 98.647 | | | |
| 7 | 0.098 | 0.887 | 99.534 | | | |
| 8 | 0.032 | 0.294 | 99.828 | | | |
| 9 | 0.017 | 0.158 | 99.986 | | | |
| 10 | 0.001 | 0.011 | 99.997 | | | |
| 11 | 0 | 0.003 | 100 | | | |

*3.3. PLS-DA and Random Forest Analysis of Oat Varieties at Different Levels of N Supply*

The supervised model PLS-DA was used to screen the agronomic traits and plant N content that could distinguish between normal and low N treatments under the two conditions. The agronomic traits and plant N content indicators were further extracted by random forest analysis, and the results showed that plant height, root–crown ratio, and plant N content, which were the top three physiological indices (importance >5), could be used as key indicators to reflect the low N tolerance of oat varieties (Figure 1A). By calculating the variable importance in projection (VIP >1, Figure 1B) of each index to quantify the ability to distinguish agronomic traits and plant N content in oat varieties, it was found that plant N content and dry weight of belowground plant parts contributed more in distinguishing plants in normal and low N treatment groups and were highly correlated with low N treatment in oat varieties.

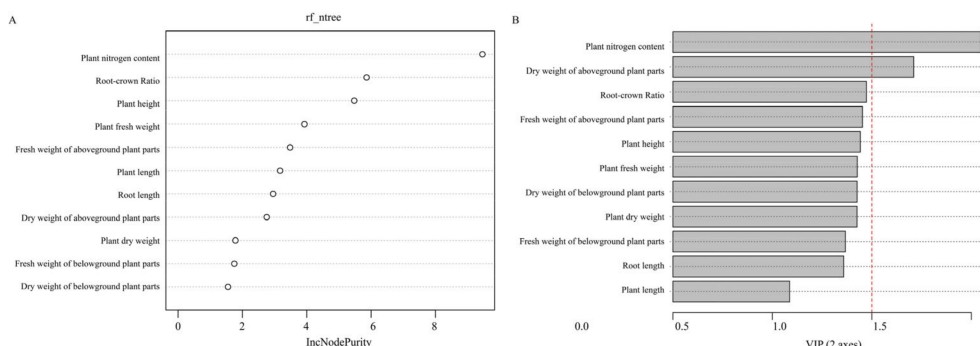

**Figure 1.** Ranking the importance of agronomic traits and plant N content indicators of normal and low N tolerance of oat varieties based on PLS-DA and random forest analysis: (**A**) VIP value ranking of PLS-DA model variables; (**B**) Importance ranking of random.

*3.4. LASSO Regression Analysis and Model Evaluation of Oat Varieties at Different Levels of N Supply*

Based on the changes in agronomic traits and plant N content of oat varieties under different N supply conditions, LASSO regression was used to screen the evaluation indicators of low N tolerance of oat varieties, and the results are shown in Figure 2. Based on the LASSO step, the minimization of the mean squared error was between 0.9 and 1.0 (Figure 2A). After cross-validation, the beta value was determined to be 8 and calculated the coefficients of the indexes were. The higher absolute values of the coefficients indicated a higher association with low N culture, and indexes with coefficients of 0 were considered to be less affected by low N stress (Figure 2B). The final selection of indicators related to low

N tolerance of oat varieties included plant height, root length, fresh weight of aboveground plant parts, dry weight of belowground plant parts, dry weight of aboveground plant parts, and plant N content. These indicators and their coefficients were constructed as a model to predict groups of the raw data, and the diagnostic performance of the model was evaluated using ROC analysis (Figure 2C). The model could predict well the distinction between normal and low N treatment groups of the 30 oat varieties (AUC = 0.912, 95% CI: 0.867–0.957). To verify that the model distinguished between plants under different N supply conditions, beeswarm plots were used to visualize the model-predicted outcomes, and the model could distinguish well between the normal and low N treatment groups with high accuracy (Figure 2D).

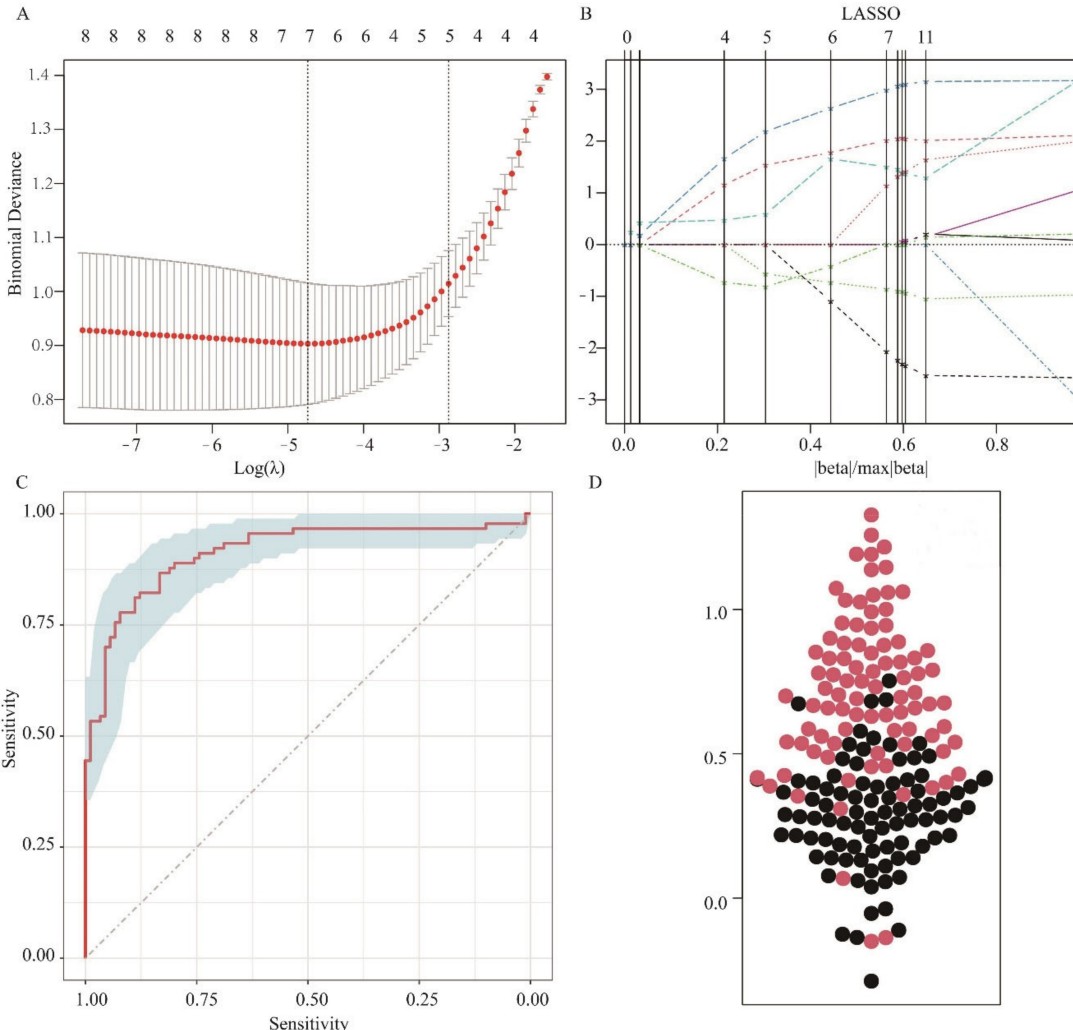

**Figure 2.** LASSO regression analysis and model evaluation of agronomic traits and plant N content indicators of oat seedlings under normal and low N treatment: (**A**) LASSO regression model convergence results; (**B**) LASSO regression distribution run results; (**C**) Model predicted ROC; (**D**) Beeswarm plot, black dots represent normal N treatment, red dots represent low N treatment.

### 3.5. Membership Function Analysis to Evaluate the Traits of Oat Seedlings at Different Levels of N Supply

This study combined CV, correlation analysis, PLS-DA, random forest analysis, and LASSO regression analysis, and used the dry weight of aboveground plant parts, root-crown ratio, and plant N content as evaluation indices. The analysis of the membership function showed the top three varieties with strong low N tolerance were Jiayan 2, Qingy-

ongjiu 035, and Qingyin 2, and the top three varieties with poor low N tolerance were Qingyongjiu 003, Qingyongjiu 021, and Qingyongjiu 016 (Table 5, Figure 3).

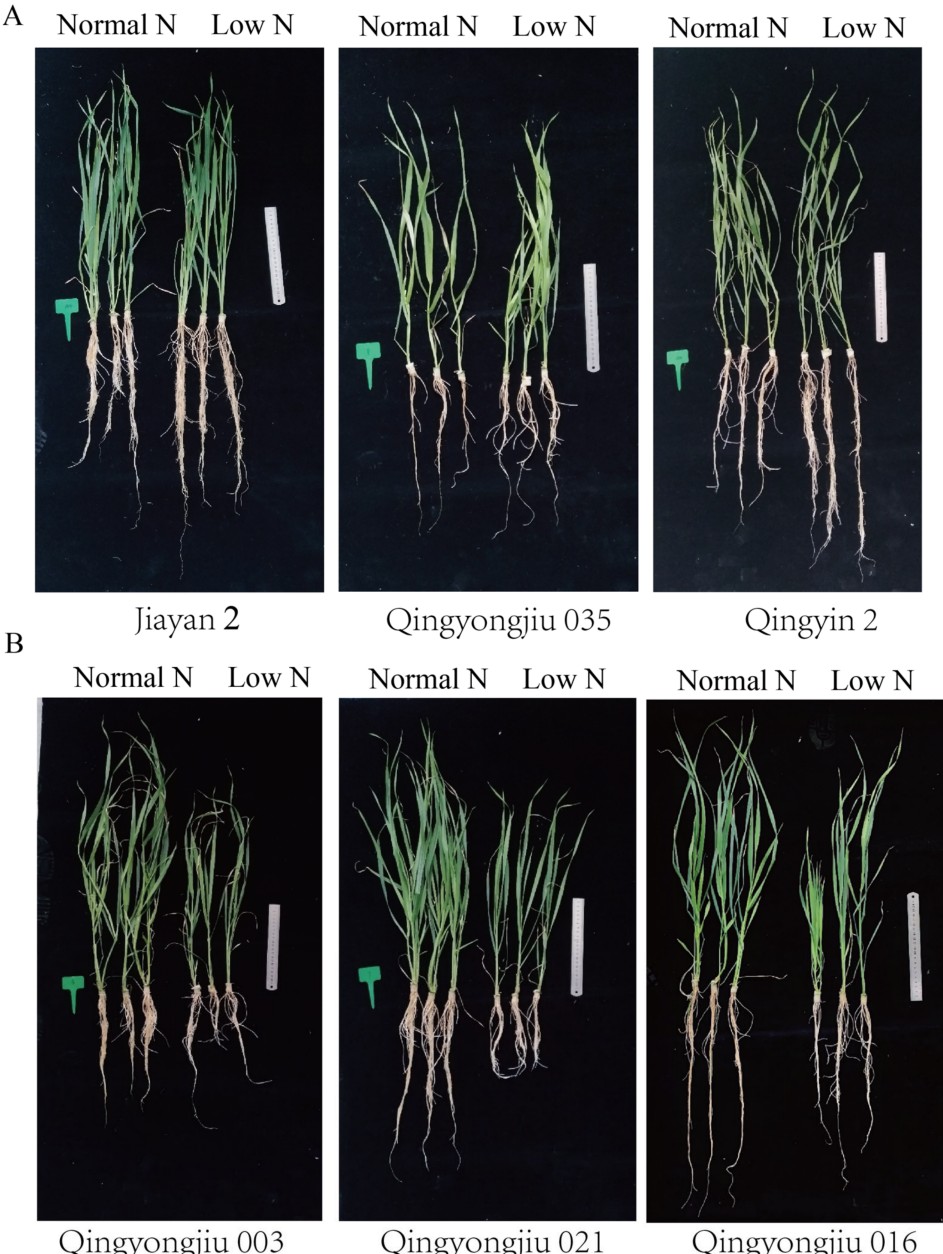

**Figure 3.** Phenotypic maps of three stronger low N-tolerant and three weaker low N-tolerant oat varieties were screened: (**A**) stronger low N-tolerant oat varieties, (**B**) weaker low N-tolerant oat varieties.

**Table 5.** Membership function analysis to evaluate the low N tolerance of oat seedlings.

| Varieties | Plant Length | Plant Height | Root Length | Plant Fresh Weight | Fresh Weight of Aboveground Plant Parts | Fresh Weight of Belowground Plant Parts | Dry Weight of Belowground Plant Parts | Dry Weight of Aboveground Plant Parts | Root-Crown Ratio | Plant N Content | Plant Dry Weight | Mean | Low N Tolerance Ranking |
|---|---|---|---|---|---|---|---|---|---|---|---|---|---|
| D1 | 0.339 | 0.174 | 0.395 | 0.189 | 0.258 | 0.123 | 0.225 | 0.293 | 0.737 | 0.515 | 0.252 | 0.318 | 24 |
| Bayan 3 | 0.394 | 0.453 | 0.312 | 0.35 | 0.395 | 0.316 | 0 | 0.446 | 1 | 0 | 0.302 | 0.361 | 21 |
| Bayan 5 | 0.229 | 0.442 | 0.115 | 0.261 | 0.23 | 0.486 | 0.409 | 0.241 | 0.347 | 0.7 | 0.23 | 0.335 | 22 |
| Qingyin 2 | 0.556 | 0.665 | 0.388 | 0.852 | 0.995 | 0.611 | 0.596 | 0.987 | 0.853 | 0.478 | 0.876 | 0.714 | 3 |
| Linna | 0.713 | 0.527 | 0.617 | 0.611 | 0.652 | 0.556 | 0.663 | 0.677 | 0.648 | 0.419 | 0.657 | 0.613 | 8 |
| Qinghai 444 | 0.901 | 0.625 | 0.849 | 0.305 | 0.29 | 0.387 | 0.494 | 0.327 | 0.364 | 0.616 | 0.333 | 0.499 | 14 |
| Qinghaitianyanmai | 0.721 | 0.454 | 0.649 | 0.525 | 0.581 | 0.449 | 0.322 | 0.636 | 0.922 | 0.689 | 0.537 | 0.589 | 10 |
| Qingyan 1 | 0.696 | 0.552 | 0.59 | 0.662 | 0.691 | 0.617 | 0.43 | 0.85 | 0.856 | 0.568 | 0.723 | 0.658 | 5 |
| Jiayan 2 | 0.956 | 1 | 0.662 | 0.824 | 0.832 | 0.788 | 0.48 | 0.992 | 0.961 | 0.749 | 0.837 | 0.825 | 1 |
| Qingyongjiu 002 | 0.558 | 0.472 | 0.53 | 0.536 | 0.419 | 0.847 | 0.641 | 0.571 | 0.46 | 1.000 | 0.567 | 0.6 | 9 |
| Qingyongjiu 003 | 0.213 | 0.066 | 0.349 | 0.069 | 0.078 | 0.141 | 0.001 | 0.095 | 0.754 | 0.541 | 0.04 | 0.213 | 28 |
| Qingyongjiu 008 | 0.26 | 0.209 | 0.395 | 0.012 | 0.054 | 0.015 | 0.059 | 0.091 | 0.528 | 0.85 | 0.049 | 0.229 | 27 |
| Qingyongjiu 021 | 0.542 | 0 | 0.204 | 0.014 | 0.054 | 0 | 0.044 | 0.128 | 0 | 0.951 | 0.083 | 0.184 | 29 |
| Qingyongjiu 016 | 0 | 0.24 | 0 | 0 | 0 | 0.069 | 0.139 | 0 | 0.145 | 0.598 | 0 | 0.108 | 30 |
| Qingyongjiu 035 | 0.605 | 0.587 | 0.488 | 1 | 1 | 0.97 | 1 | 1 | 0.596 | 0.789 | 1 | 0.821 | 2 |
| Qingyongjiu 044 | 0.154 | 0.377 | 0.096 | 0.188 | 0.164 | 0.357 | 0.113 | 0.214 | 0.652 | 0.603 | 0.164 | 0.28 | 26 |
| Qingyongjiu 045 | 0.466 | 0.551 | 0.339 | 0.697 | 0.591 | 1 | 0.623 | 0.671 | 0.652 | 0.746 | 0.629 | 0.633 | 7 |
| Qingyongjiu 055 | 0.531 | 0.592 | 0.395 | 0.652 | 0.671 | 0.626 | 0.503 | 0.898 | 0.872 | 0.865 | 0.779 | 0.671 | 4 |
| Qingyongjiu 065 | 0.266 | 0.173 | 0.323 | 0.575 | 0.585 | 0.585 | 0.476 | 0.665 | 0.832 | 0.384 | 0.589 | 0.496 | 15 |
| Qingyongjiu 067 | 0.192 | 0.324 | 0.155 | 0.218 | 0.189 | 0.354 | 0.327 | 0.286 | 0.471 | 0.809 | 0.265 | 0.326 | 23 |
| Qingyongjiu 068 | 0.371 | 0.34 | 0.342 | 0.55 | 0.533 | 0.597 | 0.466 | 0.607 | 0.717 | 0.714 | 0.567 | 0.528 | 12 |
| Qingyongjiu 086 | 0.163 | 0.183 | 0.204 | 0.247 | 0.25 | 0.288 | 0.178 | 0.286 | 0.712 | 0.652 | 0.246 | 0.31 | 25 |
| Qingyongjiu 087 | 0.594 | 0.69 | 0.413 | 0.409 | 0.52 | 0.253 | 0.229 | 0.5 | 0.891 | 0.466 | 0.412 | 0.489 | 16 |
| Qingyongjiu 091 | 0.681 | 0.551 | 0.552 | 0.249 | 0.266 | 0.26 | 0.288 | 0.3 | 0.617 | 0.375 | 0.297 | 0.403 | 20 |
| Qingyongjiu 093 | 0.758 | 0.738 | 0.561 | 0.352 | 0.441 | 0.229 | 0.383 | 0.468 | 0.69 | 0.529 | 0.433 | 0.507 | 13 |
| Qingyongjiu 096 | 0.725 | 0.43 | 0.721 | 0.313 | 0.429 | 0.139 | 0.267 | 0.452 | 0.772 | 0.354 | 0.384 | 0.453 | 18 |
| Qingyongjiu 097 | 0.673 | 0.454 | 0.602 | 0.214 | 0.257 | 0.194 | 0.187 | 0.281 | 0.704 | 0.683 | 0.234 | 0.408 | 19 |
| Qingyongjiu 112 | 1 | 0.376 | 1 | 0.529 | 0.66 | 0.361 | 0.403 | 0.807 | 0.92 | 0.425 | 0.684 | 0.651 | 6 |
| Qingyongjiu 872 | 0.674 | 0.488 | 0.597 | 0.597 | 0.605 | 0.586 | 0.404 | 0.45 | 0.7 | 0.435 | 0.415 | 0.541 | 11 |
| Qingyongjiu 028 | 0.52 | 0.647 | 0.357 | 0.353 | 0.351 | 0.381 | 0.236 | 0.401 | 0.826 | 0.611 | 0.344 | 0.457 | 17 |

Note: Values for "agronomic traits and plant N content" were determined by the membership function. The ranking is determined by the average score of the combined indicators of the membership function.

### 4. Discussion

N, applied in the form of N fertilizer, is one of the main inputs to agricultural production and is a major factor limiting crop yields [19,22]. A high consumption and low use efficiency of N fertilizer during agricultural production were the norms in the past [23]. About half of the N fertilizer can be absorbed by crops, while the rest half remains in the soil, causing environmental pollution [24]. Therefore, selecting and breeding low N-tolerant crops and reducing fertilizer use without reducing yield and quality are among the important ways to achieve sustainable crop development. Current research on N fertilizer reduction in oat varieties has focused on the analysis of agronomic traits in field trials and the response of different oat varieties to N using indicators such as N accumulation [25]. This approach is limited by many factors, such as seasonality and weather, and it is difficult to screen oat varieties for low N tolerance in large quantities. In contrast, the method of screening low N-tolerant oat varieties by hydroponics at the seedling stage is fast and effective and can greatly reduce the cost.

N is an important nutrient for plant growth, development, and agricultural production. N deficiency can induce epigenetic changes in plants [26]. In this study, the agronomic traits and total plant N content of 30 oat varieties were measured and analyzed under normal and low N treatments. Among these, the dry weight of aboveground plant parts showed the greatest variation, which could better reflect differences among varieties, and it could be tentatively concluded that the dry weight of aboveground plant parts was the evaluation criterion for oat tolerance to low N stress. Combined with the PCA data, it may be inaccurate to evaluate the low N tolerance of oat varieties based only on the relative value of the dry weight of aboveground plant parts, and further comprehensive evaluation and screening of multiple traits are needed. Therefore, PLS-DA and random forest analysis were used to show that plant height, root–crown ratio, plant N content, and dry weight of belowground plant parts play important roles in the normal and low N treatments of oat varieties. In the LASSO regression and by constructing the ROC model, the root–crown ratio had a higher diagnostic value for treatments, while few crown roots in maize had greater nitrogen uptake in low N soil [27]. Correlation analysis showed that plant height, root–crown ratio, plant N content, and dry weight of belowground plant parts were significantly correlated with a dry weight of aboveground plant parts and root-crown ratio, and plant N content showed a negative correlation. N is an essential nutrient for plant growth, and under the condition of N deficiency, plant roots elongate and lateral roots increase, thus using more soil space and N resources [28]. Therefore, the root-crown ratio can be used as a screening indicator of low N tolerance of oat varieties. The membership function analysis showed that the top three oat varieties (Jiayan 2, Qingyongjiu 035, and Qingyin 2) had a high dry weight of aboveground plant parts and plant N content under low N conditions. In contrast, the bottom three oat varieties (Qingyongjiu 003, Qingyongjiu 021, and Qingyongjiu 016) had a low dry weight of aboveground plant parts and plant N content under low N conditions. Given that the membership function values were positively correlated with the low N tolerance of oat seedlings, they could help screen low N-tolerant oat varieties. This method is similar to that used to screen low N-tolerant maize varieties at the seedling stage using D values [20]. Therefore, plant N content and dry weight of aboveground plant parts can be used as screening indicators of low N-tolerant oat varieties.

In summary, CV, correlation analysis, PCA, PLS-DA, random forest analysis, LASSO regression analysis and model evaluation, and membership function analysis were performed, and the results showed that plant N content, root-crown ratio, and dry weight of aboveground plant parts are important indicators of low N tolerance of oat varieties. We comprehensively evaluated low N tolerance to provide a reference for large-scale screening of low N-tolerant oat germplasm resources and breeding of low N-tolerant oat varieties. Thus, the present study proposes a more rigorous screening method for low N-tolerant varieties based on comprehensive indices of low N tolerance, which can serve as a reference for the screening of low N-tolerant crop varieties in the future.

## 5. Conclusions

N is important for plant growth and development, and excessive application of N fertilizer increases environmental pollution. Therefore, it is particularly important to screen low N-tolerant oat varieties. In this study, the agronomic traits and plant N content were analyzed in 30 oat varieties under normal and low N conditions, and plant N content, root-crown ratio, and dry weight of aboveground plant parts of oat varieties were found to be closely related to their ability to tolerate low N stress. This finding can provide a reference for large-scale screening and breeding of low N-tolerant oat varieties.

**Author Contributions:** Data curation, Y.W.; formal analysis, Y.W. and Q.Z.; funding acquisition, Z.J. (Zhifeng Jia) and G.L.; investigation, X.M.; methodology, Y.W. and K.L.; project administration, Z.J. (Zhifeng Jia) and G.L.; software, Y.W. and Z.J. (Zeliang Ju); writing—original draft, Y.W. writing—review and editing, Y.W., K.L., X.M. and Q.Z. All authors have read and agreed to the published version of the manuscript.

**Funding:** This research was financially supported by Key Laboratory of Superior Forage Germplasm in the Qinghai-Tibetan Plateau (2020-ZJ-Y03).

**Institutional Review Board Statement:** Not applicable.

**Informed Consent Statement:** Not applicable.

**Data Availability Statement:** All data supporting the findings of this study are included in the article.

**Conflicts of Interest:** The authors declare no conflict of interest.

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
