# Peer review of "Comprehensive Evaluation of Low Nitrogen Tolerance in Oat (Avena sativa L.) Seedlings"

_agronomy, doi:10.3390/agronomy13020604_

Round 1
Reviewer 1 Report
The manuscript titled “Screening of low nitrogen-tolerant oat varieties in alpine regions” has been reviewed. In the written, although the authors target on low nitrogen-tolerance variety screening for alpine regions, no evidence supposed that the results in growth chambers is available to the environment of alpine regions. The data procession is problematic because the well low nitrogen-tolerant oat variety should show minimal difference in high- and low-N-treatment, and all results should be based on the analysis of this difference. The final conclusion also be confusing on what kinds of agronomic trait is critical to breed well low nitrogen-tolerant oat variety. In addition, the writing is difficult to read through, and the raw material investigated should be listed out.
Author Response
Question 1: In the written, although the authors target on low nitrogen-tolerance variety screening for alpine regions, no evidence supposed that the results in growth chambers is available to the environment of alpine regions.
Response: Thanks for reviewer’s advice. I think including the phrase "in alpine regions" in the title is a little misleading, as this study was done in a hydroponic system. Therefore, amending the title to “Comprehensive evaluation of low nitrogen-tolerance in oat (Avena sativa L.) seedlings” would be significantly more compliant.
Question 2: The data procession is problematic because the well low nitrogen-tolerant oat variety should show minimal difference in high- and low-N-treatment, and all results should be based on the analysis of this difference.
Response: Thanks for reviewer’s advice. There were no data processing problems in this study. In this study, we first analyzed the low nitrogen tolerance indices for 10 agronomic traits and plant nitrogen content of 30 oat varieties under normal and low nitrogen treatments, in which the index with the highest variation in the coefficient of variation of the low nitrogen tolerance index (e.g., dry weight of above-ground plant parts) could be used as an index for screening low nitrogen-tolerant varieties. Combined with the PCA, it may be inaccurate to evaluate the low N tolerance of oat varieties based only on the relative value of the dry weight of above-ground plant parts, and further comprehensive evaluation and screening of multiple traits is needed. Therefore, through CV, correlation analysis, PCA, PLS-DA, random forest analysis, LASSO regression analysis and model evaluation, and membership function analysis, it was found that plant N content, root-crown ratio, and dry weight of above-ground plant parts are important indicators of low N tolerance in oat varieties.
Question 3: The final conclusion also be confusing on what kinds of agronomic trait is critical to breed well low nitrogen-tolerant oat variety.
Response: Thanks for reviewer’s advice. This study concludes that plant nitrogen content, root-crown ratio, and dry weight of aboveground plant parts of oat varieties are important screening indicators for low-nitrogen tolerant varieties. (line 369)
Question 4: In addition, the writing is difficult to read through, and the raw material investigated should be listed out.
Response: Thanks for reviewer’s advice. The writing has been checked by a national English speaker. We would like to thank TopEdit (www.topeditsci.com) for linguistic assistance during the preparation of this manuscript. The raw material is listed in Table 1 (line 86).

Reviewer 2 Report
The manuscript entitled “Screening of low nitrogen-tolerant oat varieties in alpine regions” deals with important problem of more efficiently usage of N fertilizers in oat production. The Introduction is sufficient, the methods are described clearly. The results are well presented however, the tables and figures should be placed in the text, close to the citation. Partially the results could be placed in methods because there are the descriptions of statistical analysis assumptions. The discussion should be extended to comparison of other studies in literature. The conclusions are drawn correctly.
1. Please emphasize the goal of the study by making the paragraph with the aim, just for clarity of the text.
2. Please explain why hydroponics was used instead of plots or pot studies.
3. Could you described more clearly what you treated as control?
4. Could you give the dosage of N used in the experiment recalculated on kg per hectare?
5. Line 148: Why Table 3? Where is table 1 and 2?
6. Tables and figures should be inserted into the main text close to their first citation and must be numbered following their number of appearance.
Author Response
The manuscript entitled “Screening of low nitrogen-tolerant oat varieties in alpine regions” deals with important problem of more efficiently usage of N fertilizers in oat production. The Introduction is sufficient, the methods are described clearly. The results are well presented however, the tables and figures should be placed in the text, close to the citation. Partially the results could be placed in methods because there are the descriptions of statistical analysis assumptions. The discussion should be extended to comparison of other studies in literature. The conclusions are drawn correctly.
Response: Thanks for reviewer’s advice. Some of the results have been incorporated into the methodology on request.
Question 1: Please emphasize the goal of the study by making the paragraph with the aim, just for clarity of the text.
Response: Thanks for reviewer’s advice. To ensure that the purpose of the study is clearer, further revise the purpose of the study in the article (line 50).
Question 2: Please explain why hydroponics was used instead of plots or pot studies.
Response: Thanks for reviewer’s advice. The use of seedling hydroponics to cultivate crops with low nitrogen treatment, characterize key agronomic traits associated with low nitrogen-tolerance, and screen for low nitrogen-tolerant crop resources is a prevalent research method nowadays(Liu et al., 2020; Miao et al., 2022). Compared to plots or pot studies, seedling hydroponics has the advantages of simple operation, a short testing period, and precisely controlled culture conditions during all experiments, making it suitable for rapid research and screening of a large number of crop resources.
Question 3: Could you described more clearly what you treated as control?
Response: Thanks for reviewer’s advice. Normal nitrogen is complete modified Hoagland’s nutrient solution as a control, referring to the study by Liu and Miao et al. (Liu et al., 2020; Miao et al., 2022)(Line 98)
Question 4: Could you give the dosage of N used in the experiment recalculated on kg per hectare?
Response: Thanks for reviewer’s advice. The mM unit is commonly used in laboratory testing. In this trial, the normal nitrogen treatment contained 10 mM NH4NO3, whereas the low nitrogen treatment contained 1.25 mM NH4NO3. (line 98)
Question 5: Line 148: Why Table 3? Where is table 1 and 2?
Response: Thanks for reviewer’s advice. Table 1 is shown in line 85 and Table 2 in line 166.
Question 6: Tables and figures should be inserted into the main text close to their first citation and must be numbered following their number of appearance.
Response: Thanks for reviewer’s advice. Figures and tables have been reinserted as required.

Reviewer 3 Report
Congratulations to the authors for producing a concise and informative paper that will hopefully be of use to the oat breeding and phenotyping community. It is nicely laid out and the findings are well presented. I just had a few comments/questions:
1. Are there any more details that can be provided about the oat genotypes in table 1 (e.g. common variety name or accession number)?
2. I think including the phrase "in alpine regions" in the title is a little misleading as this study was done in a hydroponic system. Perhaps consider replacing "in alpine regions" with "in hydroponic systems"?
3. I don't think it is necessary to provide the formula for Pearson's correlation coefficient in the paper as it is commonly used and known (the same applies for the coefficient of variation).
4. I believe the columns titled "variation" in table 2 should be replaced with the word "range". Variation is very close to the term "variance", which has a different meaning to the values presented in the table.
5. In table 2, it is not clear what statistical test for significance was used to compare means. The "A" and "B" labels also need to be defined in the table legend.
6. Units of measurement for each trait should be provided in table 2 and other tables where they are used (e.g. plant height in cm, weight in g, etc...).
7. The statistical test used for significance testing in the correlation analysis needs to be described. Were the significance tests corrected for multiple testing?
8. At line 158-160, it is suggested that "The results of the correlation analysis indicated a synergistic relationship between agronomic traits and plant N content of oat varieties, which helps explain the low N tolerance of oat varieties." I'm not sure that a synergistic relationship can be determined from the results of the correlation analysis.
9. At line 197, the LASSO figure is referred to as figure 3, but it is actually figure 2.
10. Could the authors describe in more detail the cross-validation used for their LASSO analysis in the methods?
11. The "Patents" section at the end doesn't need to be included.
12. It would be good if the authors could provide the raw data for the analysis (e.g. as an excel spreadsheet or csv file), but if that is not possible that is OK.
Author Response
Question 1: Are there any more details that can be provided about the oat genotypes in table 1 (e.g. common variety name or accession number)?
Response: Thanks for reviewer’s advice. These resources have been imported from abroad by our team of senior experts and have been included in the National Grassland Germplasm Repository. The above variety numbers are uniformly numbered by the National Grassland Germplasm Repository. Due to the age of the resources, other information cannot be verified at this time.
Question 2: I think including the phrase "in alpine regions" in the title is a little misleading as this study was done in a hydroponic system. Perhaps consider replacing "in alpine regions" with "in hydroponic systems"?
Response: Thanks for reviewer’s advice. The phrase "in alpine regions" in the title is a little misleading. The title "Comprehensive evaluation of low nitrogen-tolerance in oat (Avena sativa L.) seedlings" would be significantly more compliant.
Question 3: I don't think it is necessary to provide the formula for Pearson's correlation coefficient in the paper as it is commonly used and known (the same applies for the coefficient of variation).
Response: Thanks for reviewer’s advice. As Pearson's correlation coefficient and coefficient of variation are commonly used and known, they have been removed from the article as requested.
Question 4: I believe the columns titled "variation" in table 2 should be replaced with the word "range". Variation is very close to the term "variance", which has a different meaning to the values presented in the table.
Response: Thanks for reviewer’s advice. The columns titled "variation" in table 2 have been replaced with the word "range". (line 180)
Question 5: In table 2, it is not clear what statistical test for significance was used to compare means. The "A" and "B" labels also need to be defined in the table legend.
Response: Thanks for reviewer’s advice. In Table 2, t-tests were used to compare the significance of the means. The "A" and "B" labels have been defined in the table legend. (line 181)
Question 6: Units of measurement for each trait should be provided in table 2 and other tables where they are used (e.g. plant height in cm, weight in g, etc...).
Response: Thanks for reviewer’s advice. Units of measurement for each trait has been provided in table 2 and other tables where they are used. (line 180)
Question 7: The statistical test used for significance testing in the correlation analysis needs to be described. Were the significance tests corrected for multiple testing?
Response: Thanks for reviewer’s advice. This study uses Pearson's correlation coefficient for a two-tailed test of traits.
Question 8: At line 158-160, it is suggested that "The results of the correlation analysis indicated a synergistic relationship between agronomic traits and plant N content of oat varieties, which helps explain the low N tolerance of oat varieties." I'm not sure that a synergistic relationship can be determined from the results of the correlation analysis.
Response: Thanks for reviewer’s advice. The correlation analysis results do not indicate a synergistic relationship and should be appropriate relationship. (line 199)
Question 9: At line 197, the LASSO figure is referred to as figure 3, but it is actually figure 2.
Response: Thanks for reviewer’s advice. The LASSO figure is referred to as figure 2 and has been modified in the article. (line 249)
Question 10: Could the authors describe in more detail the cross-validation used for their LASSO analysis in the methods?
Response: Thanks for reviewer’s advice. A more detailed description of the methods used for their LASSO analysis has been added as requested.
Question 11: The "Patents" section at the end doesn't need to be included.
Response: Thanks for reviewer’s advice. The "Patents" section at the end has been removed.
Question 12: It would be good if the authors could provide the raw data for the analysis (e.g. as an excel spreadsheet or csv file), but if that is not possible that is OK.
Response: Thanks for reviewer’s advice. I could provide the raw data for the analysis. (See the appendix for details)
